# Online stroke forum as source of data for qualitative research: insights from a comparison with patients' interviews

James Jamison,[1] Stephen Sutton,[1] Jonathan Mant,[1] Anna De Simoni[2]

## ABSTRACT

**Objective** To determine the appropriateness of an online forum compared with face-to-face interviews as a source of data for qualitative research on adherence to secondary prevention medications after stroke.

**Design** A comparison of attributes of two data sources, interviews and a forum, using realistic evaluation; a comparison of themes around adherence according to the Perceptions and Practicalities Approach (PAPA) framework.

**Setting** Interviews were conducted in UK GP practices in 2013 and 2014; online posts were written by UK stroke survivors and family members taking part in the online forum of the Stroke Association between 2004 and 2011.

**Participants** 42 interview participants: 28 stroke survivors (age range 61–92 years) and 14 caregivers (85% spouses). 84 online forum participants: 49 stroke survivors (age range 32–72 years) and 33 caregivers (60% sons/ daughters).

**Results** 10 attributes were identified within the two data sources and categorised under three domains (context, mechanisms and outcomes). Participants' characteristics of forum users were often missing. Most forum participants had experienced a stroke within the previous 12 months, while interviewees had done so 1–5 years previously. All interview themes could be matched with corresponding themes from the forum. The forum yielded three additional themes: influence of bad press on taking statins, criticisms of clinicians' prescribing practices and caregiver burden in assisting with medications and being advocates for survivors with healthcare professionals.

**Conclusions** An online forum is an appropriate source of data for qualitative research on patients' and caregivers' issues with adherence to secondary prevention stroke medications and may offer additional insights compared with interviews, which can be attributed to differences in the approach to data collection.

[1]Primary Care Unit, Institute of Public Health, Forvie Site, University of Cambridge School of Clinical Medicine, Cambridge, UK
[2]Centre for Primary Care and Public Health, Barts and the London School of Medicine and Dentistry, Queen Mary University of London, London, UK

**Correspondence to**
James Jamison;
jj285@medschl.cam.ac.uk

### Strengths and limitations of this study

► Informed by the literature and using a realistic evaluation approach, this study provides a structured framework to systematically explore differences between two data collection approaches, which may be of use to other researchers.
► The differences in study participants and themes highlighted here could be used to inform researchers on which data collection approach would be most appropriate for the research question being asked.
► Understanding of themes was not affected by potential limitations of an online forum such as lack of knowledge of participant characteristics, absence of face-to-face interaction and inability to directly question participants.
► With no control over the direction of discussions, the effectiveness of the online forum as a source of data will depend on how well the forum posts address the research question.
► Younger forum participants were potentially more engaged with managing their condition than their offline interviewed counterparts, perhaps representing people with a good understanding of their health.

offering access to hard-to-reach groups who are often excluded (or exclude themselves) from traditional research studies.[8 9]

Internet use across the UK has grown considerably. A recent report on internet usage found that in the first quarter of 2017, approximately 89% of adults used the internet. From 2011 to 2016, internet use has risen from 52% to 78% in 65–74 years old.[10] Forty-two per cent of those 75 years and older of all genders are now internet users.[11] It is estimated that one in four people with a chronic condition who use the internet go online to find others with similar health concerns.[12] Patients engage with the internet to access health information[13] and manage chronic illness.[14]

Patients are becoming more informed about their health through using the internet.[15 16] Around 70% of Europeans who

## INTRODUCTION

In recent years, participation in online forums has increased dramatically out of a need for patients to know more about the healthcare conditions they face.[1–5] With the use of online health forums steadily increasing, greater efforts are being made to understand this mode of data collection for qualitative research.[6 7] Online patient communities represent an important source of information,

access the internet use it to obtain health information.[17] In the UK, digital technology has recently emerged as a key vehicle for the delivery of health and social care.[18] A review of the use of technology in healthcare confirmed that social media were increasingly used to communicate health information among public, patients and health professionals.[19]

As a method of capturing data on health attitudes and behaviour, the online forum offers considerable advantages, including access to large numbers of prospective participants with the potential for open and honest discussions.[8 9 20 21] Such forums have been used previously in healthcare research[22] across a range of health domains.[23–26]

In the face of increased technological change, there is a growing need to understand the potential for online sources of qualitative data[27] and their advantages and disadvantages compared with traditional data collection techniques.[28 29]

The face-to-face interview is an important qualitative data collection technique widely used in healthcare research.[30 31] This method permits close observation of respondents, flexibility to adapt the direction of conversation and the scrutiny of physical cues such as body language.

There is a growing body of literature exploring the potential for online forums, as a source of data collection compared with traditional qualitative techniques as well as a greater understanding around using each of these methods. Nevertheless, difficulties verifying participants' identity and medical condition (ie, are forum users real patients with stroke?) as well as the inability to interpret visual cues and seek clarification to questions, suggests that it may be necessary to confirm results with more established data sources, such as qualitative interviews. Confirmation of data may be deemed necessary in order to explore whether forum findings are representative of issues experienced by patients with stroke, and whether differences between the two sources could be used to decide which is better suited to addressing a particular research question. In an investigation comparing an online forum with qualitative interviews among patients with cancer, the authors concluded that the forum offered useful data for qualitative health research.[32] Similarly, comparison between an online forum and face-to-face focus groups in people with multiple sclerosis concluded that forum results were comparable.[33] Comparison of characteristics of online versus face-to-face approaches have been reported more frequently with respect to focus groups.[34 35]

Drawing on a realistic evaluation perspective,[36] the objective of the present study was to explore differences between two approaches to data collection, interviews[37] and an online stroke forum[38] by seeking to understand the attributes that underpin each data source, exploring the context within which each data collection occurs and comparing barriers and facilitators to adherence to secondary prevention medication classified thematically according to the Perceptions and Practicalities Approach (PAPA)[39] framework. Our overall aim was to offer a structured way to systematically explore the differences between these two data collection approaches and highlight the characteristics of an online stroke forum as a source of data for qualitative research, which may be of use to other researchers.

## METHODS

### Design

Comparison of themes around barriers and facilitators of adherence to secondary prevention medications after stroke in two independent studies, qualitative semistructured interviews and an online forum. Although one author (JJ) initially coded the data in both studies, a subset of each dataset was double coded by a different author in each of the studies; therefore, thematic analyses were independently validated. As interviews are a widely accepted method in qualitative research,[40] we used this as the standard against which to compare forum data. Differences and similarities in the data were examined, and results were compared and contrasted to explore the potential of the online forum as a data collection source.

### Interview dataset

Interview participants included stroke survivors recruited through five general practice surgeries in Eastern England, along with their caregivers, as described previously.[37] In brief, purposive sampling was undertaken; patients were approached by letter, and positive responders were contacted to confirm attendance. All interviews were guided by a topic schedule, with written consent. They were conducted in the stroke survivors' own houses together with caregivers and lasted approximately 1 hour. Twenty per cent of the interviews were double coded by another author to ensure rigour and strengthen the validity of findings.

### Online forum dataset

Methods are reported in greater detail elsewhere.[38] Briefly, the source of data was the archive file of an online forum, Talkstroke, hosted by the UK charity Stroke Association, between 2004 and 2011. This was a moderated forum, set up as part of the charity website with the scope of facilitating online communication between stroke survivors and caregivers, sharing information on any aspect of stroke and offering emotional support.

Barriers and facilitators of adherence were identified through analysis of a set of predefined keywords related to secondary prevention and stroke. Forum posts were explored using thematic analysis. Key themes were developed, representing barriers and facilitators of medication adherence. As these themes were further refined, subthemes were identified, and a coding framework was developed. Forum posts were coded to identify practical and perceptual factors affecting adherence to medication, guided by the PAPA framework. In the final stage

of the analysis, themes that were identified were mapped onto the theory and then subdivided to represent barriers or facilitators of adherence. To ensure rigour, another researcher who was not involved in coding the interviews, double coded half of all the forum posts identified.

In respect to ethical approval, the Stroke Association granted permission to use the stroke forum data for research purposes before analysis of the data commenced. Informed consent was not sought from forum participants although forum users were aware that by participating in a public forum, their responses were available for others to view online. Verbatim quotes posted in the online forum were not used to protect the identity and intellectual property of participants; despite this being normal practice in qualitative research, we only used descriptions of quotes throughout the text.[9 21] To minimise the risk of interpretation bias, we ensured the paraphrasing of text reflected as closely as possible to the original forum posts. The ethical aspects of conducting research on this forum have been discussed more extensively elsewhere.[21 41]

### Procedure and analysis
To allow us to directly compare the analyses from these interviews and an online stroke forum,[38] transcripts of the interview study were reanalysed in NVivo V.10 using a thematic analysis[42] according to the PAPA framework (see details in the section Procedure and Analysis).[39 43]

### Comparison of attributes of the two data sources
The literature was explored by two authors (JJ and ADS) to isolate characteristics associated with face-to-face and online forum approaches to qualitative data collection.[6 7 32 34 35] The evidence was discussed with experts in qualitative research methods, and subsequently 10 attributes were identified as representing the key characteristics of both methods of data collection. Attributes were categorised according to the domains of realistic evaluation: context, mechanisms and outcomes. The classification of attributes into context, mechanisms and outcomes was discussed until a final consensus was reached. The key attributes that were considered to represent important aspects of each data source were subsequently applied in the context of collecting/interpreting research data from each source.[36] Context included the attributes location and sampling; mechanisms included the attributes participation, dynamic of interaction, contribution, timing, guidance and communication; and outcomes included activities and reporting (see table 1).

### Comparison of themes using PAPA thematic analysis
Key themes arising from the data were classified according to the PAPA framework[39] and interpreted according to the following two categories of the PAPA framework.

#### Perceptions: necessity beliefs and concerns
Perceptual barriers and facilitators of medication adherence in stroke survivors and caregivers are explored within both sources of data, according to their classifications as necessity beliefs, that is, doubts about personal need for medication to maintain or improve current and future health and concerns about secondary prevention treatment.

#### Practicalities: capability and resources
Barriers and facilitators that stroke survivors and caregivers face around their capability of taking/giving medication and the resources available to undertake such behaviour.

### RESULTS
Details of participant characteristics are reported in table 2. Of the 42 interviewed participants, two-thirds were stroke survivors and one-third were caregivers. The median age of interviewed survivors was 72 years (range: 61–92 years), and the majority were female (21/28; 75%). The majority of interviewed caregivers were the stroke survivor's spouses. Sixty-four per cent of stroke survivors experienced a stroke within the previous 5 years, 22% in the last 12 months. Fifty per cent suffered from a stroke, predominantly ischaemic, while 50% suffered from a TIA. Interview participants were recruited from a single UK region.

Of the 84 online users, 58% were stroke survivors and 39% were caregivers. Forum survivors were on average aged 50 years (range: 32–72 years). The median age of stroke survivors talked about by caregivers on the forum was 66 years (range: 46–91 years), and 57% were female. Sixty-two per cent of caregivers in the online forum were daughters or sons, 28% spouses and the remainder was family members such as siblings or in-laws. Ninety per cent of forum participants, who reported time since stroke, experienced it within the previous 5 years, while 53% experienced it within the previous 12 months. It was not possible to determine the type of stroke experienced by users in the online forum.

A small number of participants on the forum were prolific users, commenting frequently and offering encouragement to other participants. Forum users came from all over the UK.

Despite the differences between the two sources of data as highlighted in table 1, all key themes about barriers and facilitators to adherence to secondary prevention medications that emerged from the interview study could be matched with corresponding themes from the online forum (see table 3). The comparison of themes in the two data sources was facilitated by their classifications according to the PAPA framework, and details are reported in table 3.

Three additional themes were identified in the forum, which did not emerge from interviews. First, stroke survivors openly discussed the influence of bad press on medication taking, in particularly around statins. Second, forum users raised concerns around healthcare professionals' prescribing practices and financial incentives to prescribe. Third, caregivers' difficulties in ensuring adherence to secondary prevention medications and

**Table 1** Key differences between an online forum approach and an interview approach to data collection

| Attribute | Online forum<br>Recognised feature identified in the literature<br>versus *features identified in the present study* | Semistructured interviews<br>Recognised feature identified in the literature<br>versus<br>*features identified in the present study* |
|---|---|---|
| **Context** | | |
| Location | Respondents from across a wide geographical area can participate at own convenience.[54] | Interviewees geographically restricted. |
| | *Patients' own home: UK.* | *Patient's own home: East of England.* |
| Sampling | Voluntary participation/self-selection. Recruitment does not require collaboration between clinical sites or support of professional staff.[55] | Purposive recruitment in healthcare settings guided by sampling techniques based on population demographics including age, gender and disability. |
| | *Voluntary self-selection by participant: no sampling criteria, no restriction on age. Verification of stroke/TIA diagnosis not possible. Most participants taking part within the first 5 years since the cerebrovascular event.* | *Purposive sampling: GP screened, predefined criteria to achieve maximum spread of gender and disability.*<br>*Age: 55 years and over. Confirmed stroke/TIA diagnosis.* |
| **Mechanisms** | | |
| Participation | *Multiple participants per conversation thread: stroke survivors or caregivers. Conversation possible between survivors, survivors and caregivers or caregivers with other caregivers.* | *Maximum of two or three participants per single interview conversation: researcher, stroke survivor and caregiver.* |
| Dynamic of interaction | Discussion conducted remotely. Relative anonymity can encourage users to feel uninhibited.[5 56]<br>Likelihood of expressing honest opinions about sensitive issues. | Engagement can be actively encouraged. Face-to-face approach enables development of rapport between interviewer and participant.[57] |
| | *Less knowledge of participants and participants remain anonymous. No influence of researcher on participation.* | *Researchers gain knowledge of interviewees, development of researcher–participant rapport, active encouragement of participation.* |
| Response contribution | Longer conversations allow for a broader understanding of the subject matter and potential for greater depth.[58] Discussion threads generate reflection and greater description among users.[33] | Probing questions from researcher seeking clarification or to pursue a more detailed response.[40] |
| | *Potential for significant individual contribution. Responses shaped by other peer contributors. More opportunity for self-reflection. More frequent comments.* | *Maximum of two individual contributions (patients and caregiver).*<br>*Responses shaped by researcher interaction. Less opportunity for self-reflection. In-depth comments encouraged by researcher.* |
| Timing of event | Users can post repeatedly and frequently on many topics over a long period of time. | Interview is a single event occurring at one point in time. Maximum of two contributors to the interview discussion. |
| | *Single or multiple participation over time. Ability to contribute to discussion on multiple occasions/topics.* | *Single participatory event. Contribution fixed to a single time period.* |

**Table 1** Continued

| Attribute | Online forum<br>Recognised feature identified in the literature<br>versus *features identified in the present study* | Semistructured interviews<br>Recognised feature identified in the literature<br>versus<br>*features identified in the present study* |
|---|---|---|
| Guidance | User freedom to choose what to discuss and how frequently to contribute to free-flowing discussion threads.[5]<br>Posts created through peer-to-peer communication, without professionals' involvement and influence.[59]<br>Response shaped by contribution of other survivors or caregivers.<br><br>*Free or peer-guided discussions.* | Follows a predefined line of questioning.<br>Several key questions define the area to be explored.[40]<br>Researcher oversees the direction of conversation.<br><br>*Guided conversation: responses to predefined questions in topic guide.* |
| Communication | Permits broad accessibility and asynchronicity with online communication.[5 60]<br>Restricted to those with internet access.[34]<br><br>*Indirect communication, via computer, no physical proximity, asynchronous.* | Direct face-to-face, synchronous communication.<br><br><br>*Direct communication, face to face, synchronous.* |
| **Outcomes** | | |
| Activities | No physical transcription is required; user contributions printed automatically, improving credibility of data.[7]<br>Potential for inaccurate interpretation through misunderstanding nuances in the data may still exist.[7]<br><br>*No audio recording.*<br>*Automatic transcription printed directly from forum*<br>*No field notes.* | Transcription is key to representing the individual and dependability of data.<br>Transcription opens data to misinterpretation or misunderstanding.[61]<br>Rigour and accuracy in transcribing is integral to the analysis process, influencing the degree of dependability of data.[61]<br><br>*Interviews are audio recorded and interviews are transcribed. Potential for ambiguity through inaccurate transcribing*<br>*Field notes taken during interviews* |
| Reporting | Forum posts are moderated before appearing online, effect on the data collected is relatively unknown. Moderation processes can influence engagement in online communities.[62]<br>*Third-party moderation leading to possible exclusion of data.* | <br><br><br><br>*No exclusion of data prior to analysis.* |

Location: geographical area of the research. Sampling: sampling method used to recruit participants. Participation: individuals participating in conversations. Dynamic of interaction: knowledge of participant determined by level of engagement. Response contribution: level of contribution to the conversation by individuals. Timing of event: frequency of participation over time. Guidance: level of conversation guidance and level of freedom to discuss. Communication: face to face versus distance communication. Activities: need for audio recording and transcription activities. Reporting: moderation of data before analysis.
GP, General Practitioner; TIA, Transient ischaemic attack.

struggles in acting as advocates for patients with healthcare professionals.

To understand attributes in the context of data source, we examined how they related to the themes identified within the two studies. Table 1 also shows the description of the two sources according to the data collection attributes identified.

The forum facilitated access to participants' views from across a wide geographical area compared with the views of a small group of survivors within a specific context (ie, Eastern England GP practices). This may have limited interview participant's views, with the forum drawing on more varied and wide-ranging healthcare setting experiences.

The sampling of participants in the forum was reflected in the theme '*How seriously people take medicine for secondary prevention of stroke*', with online users being familiar with negative press on statins from across a variety of information sources, including research papers and online sources. Older interview participants recruited through GP practices instead might not be able to access this as easily, as reported in the '*Knowledge of stroke and medications*' theme. Within this theme, interviewed stroke survivors reported looking for medicine information in

**Table 2** Characteristics of participants of the online forum and interview study

| Sample characteristics | Interviews N (Median) (Range) | Online forum N (Median) (Range) |
|---|---|---|
| Total participants | 42 | 84 |
| **Age** | | |
| Survivor | 72 (61–93) | 50 (32–72) |
| Caregiver* | – | |
| **Gender** | | |
| Male – survivor | 7 | 20 |
| Female – survivor | 21 | 26 |
| Not known – survivor | – | 3 |
| Caregiver of male survivor* | 4 | 20 |
| Caregiver of female survivor* | 10 | 12 |
| Unknown gender and unknown identity | – | 3 |
| **Identity person posting** | | |
| Stroke survivor | 28 | 49 |
| Caregiver | 14 | 33 |
| Not known | – | 2 |
| **Years since stroke** | | |
| 0–12 months | 4 | 37 |
| 1–5 | 14 | 25 |
| 6–10 | 6 | 4 |
| 11–15 | 2 | 2 |
| 15+ | 2 | 1 |
| Unknown | – | 15 |
| **Type of stroke** | | |
| TIA | 14 | – |
| Ischaemic stroke | 13 | – |
| Haemorrhagic stroke | 1 | – |
| **Caregiver identity** | | |
| Daughter/son | 2 | 20 |
| Spouse | 12 | 9 |
| Other (in-law/sister) | – | 3 |
| Unknown | – | 1 |
| **Number of posts about secondary prevention** | | |
| | | 37 (1 participant) |
| | | 15 (1 participant) |
| | | 1 (44 participants) |
| | | 2 (19 participants) |
| | | 3 (6 participants) |

Continued

**Table 2** Continued

| Sample characteristics | Interviews N (Median) (Range) | Online forum N (Median) (Range) |
|---|---|---|

*Refers to 'caregiver' in interviews and 'Patient talked about by caregiver' in the forum discussions.
TIA, Transient ischaemic attack.

leaflets inside drug boxes, with forum participants seeking information mainly from resources such as healthcare professionals or online peers. Forum participants seemed keener to adopt a joint approach for medication taking with health professionals, which included patient–clinician shared decision making about stopping medications.

Participation in the online forum meant that users had the possibility of taking part in multiple discussion threads. Despite the inability to ask clarification questions or to probe participants of the forum, survivors and caregivers could read and reply to each other's posts in an asynchronous way, with online discussions allowing an in-depth exploration of themes about barriers and facilitators of secondary prevention medication taking. The '*ability to self-care*' theme is an example: survivor's perspective of handing over all responsibility of medications to caregivers was enriched by caregivers' posts describing how they were acting at times as advocates for stroke survivors with health professionals. As has been highlighted in previous research in the field,[33] the caregiver/survivor discussion dynamic among forum participants permitted conversations across and within patient and caregiver groups. Stroke survivors who were forum users offered advice and suggestions about medication as well as seeking reassurance and support. At the same time they were providing advice to caregivers on medicine taking or dealing with medication refusal, as reported in the themes on *regimen complexity and burden of treatment*.

The relative anonymity during online forum discussion and the absence of researcher's influence favoured openness among forum participants. For example, forum users were more likely to make frank admissions about decisions to refuse medicines, particularly statins, and about GP role in advising on medicines, as shown in the '*Taking medication*' theme. While the younger age of forum participants may have contributed to this, interviews with stroke survivors were conducted in the presence of the caregiver, and this may have encouraged a level of self-censorship. Discussions around this theme went as far as including clinicians' financial motivation behind prescriptions and questioning whether this was prioritised over health benefit. This contrasted with the dynamic reported by interview participants who seemed to rely on GPs and were willing to do as the GP said.

Forum users had multiple opportunities for participation, with an open line of questioning guided by other survivors developing the conversation and widening the scope of the discussion. This contributed to data richness

**Table 3** Comparison of themes identified through analysis of semistructured interviews and the online forum

| Perceptions | | | | |
|---|---|---|---|---|
| Key themes identified (interviews) | Interview quote* | Interview participant (Gender, age, participant ID, stroke type) | Forum quote† | Forum participant‡ (Gender, age, participant ID) |
| **1. Knowledge of stroke and medications** | **Treatment necessity** | | **Treatment necessity** | |
| | The importance of taking this exactly on time is trivial. I would probably survive for a week, if I didn't take the. For a month I'd probably survive. It would not make any difference in two days. | Male, 86 years, N.03, ischaemic stroke | A female survivor commented that it was better to take a few extra tablets from the GP than to experience a stroke. Tablets were provided to prevent a further stroke, and she stressed that they shouldn't be stopped except on professional advice | Female, age 51 years, age at stroke 51 years, N.17 |
| | Whenever I've got a new pill or anything I'd read the instructions only because they've made a mistake before now, like for instance they gave me one which I'm allergic to... So I keep check of what I'm taking now. | Male, 80 years N.04, TIA | A stroke survivor recalled being on 75 mg of aspirin as well as beta blockers, however, his nephew who was a consultant surgeon, suggested that had he been taking warfarin instead of the aspirin he may not have suffered a second stroke. *(Bad press influences attitudes to medicines)* | Male, age 67 years, age at stroke 55 years, N.82 *Female, age 56 years, age at stroke 56 years, N.66* Female, age unknown, age at stroke unknown, N.74 |
| | | | *A female survivor read about the hype around statins and stated she still didn't have confidence in them. She had read a research paper on statins suggesting they only added an extra 9 months of life. Her mum had been taking statins for 14 years and this still didn't prevent her arteries from clogging up.* *(Bad press influences attitudes to medicines)* | |
| | | | A survivor who had suffered 2 mini-strokes and been prescribed aspirin and cholesterol lowering statin but was refusing to take them because she had read in the press about bad side effects that they caused. | |
| **2. Doubts about medicines** | I think aspirins are good for you. That's the one I fancy. Well it thins the blood and the blood it flows and that stops any clots so I do like to take it. I just don't see why I'm taking the other medication. I'm not fat or anything like that. I don't get very high blood pressure and well cholesterol, what is cholesterol? | Male, 75 years, N.24, ischaemic stroke | A survivor acknowledged statins were used to control cholesterol, but questioned whether high cholesterol was actually a problem. He believed strokes occurred frequently, regardless of cholesterol levels. He talked about the 'Cholesterol Myth' having researched the topic on the Internet. He said he was feeling confused about the value of statins and taking these when in reality they weren't needed. | Male, age 67 years, age at stroke 55 years, N.70 |
| **3. Realisation of the importance of medicines** *Differing attitudes to medicines* | At one time I wouldn't take a pill, I wouldn't even take an aspirin. Now I take it because I understand it keeps me alive. I just think it's fate, that's the way I look at it. If I stop taking medication I might as well lie down in the fast lane. | Male, 67 years, N.12, ischaemic stroke | A male survivor already suffered 2 strokes and said it was impossible to ever fully recover from the experience. He said after his first stroke he was prescribed tablets he didn't take and he realizes this was a big mistake. | Male, age 67 years, age at stroke 55 years, N.82 |
| | Well I don't know what I'd be without taking them put it that way... because I've had a stroke and I've been fortunate. | Male, 73 years, N.11, ischaemic stroke | A female survivor felt it was better to take tablets from the GP than to experience another stroke. Tablets were provided to prevent another stroke and shouldn't be stopped except on professional advice. | Female, age 51 years, age at stroke 51 years, N.17 Female, age 52 years, age at stroke 52 years, N.76 |
| | They keep me going, keep me on the straight and narrow. I refused it and I said well it's not because its rat poison. If you tell me I've got warfarin, I must be ill but if I take aspirin I can't be that ill. | Female, 71 years, N.22, TIA | Another survivor remarked that, although being on pills was an inconvenience and she had stopped some medication, she continued to take aspirin and statin which she considered important. | Female, age 42 years, age at stroke 42 years, N.35 |
| | The only thing I like, I think aspirin's good for you, that's the only one I fancy. Well it thins the blood, and well thinning the blood makes it flow better and that stops any clots so I do like, like to take it. | Male, 75 years N.25, ischaemic stroke | A survivor had suffered 2 strokes in the previous year, but none since commencing warfarin. She felt reassured by taking warfarin and worried about coming off the medication. | Male, age 35 years, age at stroke 34 years, N.71 |
| | I just take them because the hospital prescribed them. If the doctor prescribed them I probably wouldn't bother. I'd probably say forget about it. He's a consultant so he should know what he's talking about. | Male, 67 years, N.15, ischaemic stroke | A survivor described how he trusted his vascular surgeon who had changed his medication from warfarin to aspirin and statin. The survivor was happy to take aspirin and felt it would be good to continue as the surgeon also took it regularly, concluding it must be beneficial and would enable him to live longer. | Female, age 52 years, age at stroke 52 years, N.76 |
| | | | *A female survivor decided to reduce cholesterol using diet instead, because of side effects from medication. She felt that once the symptoms completely disappeared she wouldn't take a statin again. She said she would start taking olive oil and follow a healthy diet to keep her cholesterol balanced naturally. She said she would continue aspirin as it didn't seem to cause any side effects.* | |

Continued

**Table 3** Continued

| Perceptions | | | | |
|---|---|---|---|---|
| Key themes identified (interviews) | Interview quote* | Interview participant (Gender, age, participant ID, stroke type) | Forum quote† | Forum participant‡ (Gender, age, participant ID) |
| | Treatment concerns | | Treatment concerns | |
| **4. How seriously people take medicines for secondary prevention of stroke** | I wouldn't take them because I still, to me, blood pressure and cholesterol tablets to me I don't see what they're doing for me. Well she gave me blood pressure pills and that but I didn't take them. I felt so, I didn't bother, pity really. But never mind. I do now. I'm religious about that. I'll have another stroke if I don't. Didn't want to put the family through that again. | Male, 75 years, N.24, ischaemic stroke, Male, 65 years, N.02, ischaemic stroke | A female survivor who had read bad reports about statins reported being nervous about them. She didn't want to jeopardise feeling good by taking medication that she wasn't convinced she needed. A survivor refused statins after her first stroke because of side effects. However, after suffering a second one she was now worried enough to take them. | Female, age 54 years, age at stroke 54 years, N.37 Survivor, female, age 68 years, age at stroke 67 years, N.14 |
| **5. Taking medications** *Non-adherence to medicine* *Trust in GP* | Well now and again I forget the cholesterol because that's the one at night and it's the only one I take at night. So if the doctor says take ten pills a day, I'll, I'll do it.... he makes the decision and erm he, he's the boss man as you might say, who knows what he's up to. I do exactly as the Dr tells me. He's the Dr isn't he. He should know better than what I do. I don't push them anymore and say well you know I don't like taking this. | Male, 67 years, N.15 ischaemic stroke, Male, 87 years, N.8, TIA, Male, 80 years N.04, TIA | A male survivor said he was on 2 tablets for blood pressure and that he continued to take one every day. But the other was a diuretic and having got fed up frequently running to the toilet, he decided to check his blood pressure every day and would skip the diuretic if it was fine. A male survivor agreed with his doctor to stop taking a blood pressure tablet because of intolerable side effects, and his wife being a nurse made it easier. He felt strongly that doctors are there to advise not instruct. (Collaborating with GP/patient) A caregiver said that her husband ceased taking medications except aspirin, because of side effects. He made this decision, together with the GP and stressed the importance of doing this before stopping tablets. Concern (Prescribing concerns) A caregiver (sister) suggested that GPs shouldn't be paid for prescribing statins with the decision based on clinical judgement alone. The involvement of money could lead to medication being over prescribed for financial reasons. Concern (Prescribing concerns) A survivor described taking 9 pills a day for stroke and its side effects and felt that the GP should understand which were necessary. Following another appointment her consultant was furious about the medications she had been prescribed. | Male, age unknown, age at stroke unknown, N.63, *Male, age unknown, age at stroke unknown, N.63* Male, age 54 years, age at stroke 52 years, N.68 Gender and age unknown, age at stroke unknown, N.78 Female, age 37 years, age at stroke 36 years, N.41 |
| **Practicalities** | | | | |
| | Capability/resources | | Capability/resources | |
| **6. Ability to self-care** | My wife sorts it out and that's why I don't know so much about it you see she [taps].She puts them there, I take them and that's it. | Male, 80 years, N.04, TIA | A caregiver stated that she was providing the stroke survivor with all of his medication due to his poor memory as a result of the stroke. She was now in complete control of his medication which she was happy about but it was difficult as he was a loved one and something she had no training for. (Caregiver as an advocate for the stroke survivor) A female caregiver described consistently trying to have her husband's 40 mg statin dosage reduced by his GP. As a result of the high dosage he was chronically tired, so he stopped taking statins. (Caregiver as an advocate for the stroke survivor) A caregiver recommended being firm with GPs about being prescribed atorvastatin if simvastatin was not tolerated, as atorvastatin was a bit more expensive but recommended by NICE guidelines as an alternative. | Female, age 46 years, age at stroke 40 years, N.5 Male, age 54 years, age at stroke 52 years, N.68 Gender and age unknown, age at stroke unknown, N.18 |

Continued

**Table 3** Continued

Perceptions

| Key themes identified (interviews) | Interview quote* | Interview participant (Gender, age, participant ID, stroke type) | Forum quote† | Forum participant‡ (Gender, age, participant ID) |
|---|---|---|---|---|
| **7. Taking medication**<br>*Problems swallowing*<br>*Accessing packaged medicines* | The big ones, I, do actually feel I have to swallow two or three times to get them down.<br>Some of the, the pills are a hell of a trouble, you know the bubble wrap, flipping them out especially with my hands not as strong as they should be.<br>I'd have to rely on the wife to…cause I can't get them out the packet, just just can't get your hands in. | Male, 66 years, N.10, TIA<br><br>Male, 87 years, N.08, TIA<br><br>Male, 65 years, N.02, ischaemic stroke | A male survivor described 'swallow panic', that is, fear of choking when trying to take Dipyridamol capsules. The user said it took around 3 months before he got over that.<br>A caregiver (son in law) mentioned that despite the use of a nomad tray, tablets were still being taken from the wrong day with several days tablets being taken in a single day. The stroke survivor often didn't take the time to work out the days or to look at the calendar.<br>A survivor agreed with another user about the problem with the size of dipyridamole tablets, which were getting stuck in the pill box organizer. | Male, age unknown, age at stroke unknown, N.85<br>Male, age unknown, age at stroke unknown, N.40<br>Female, age 46 years, age at stroke 45 years, N.30 |
| **8. Medication routines** | I only remember to take the others if I take them out of the cupboard the night before and leave them on the top. If I didn't I would probably forget…. because it isn't the first thing that I think of.<br>I usually take it around 5 o'clock which strangely enough is about the time that we feed the dog and normally speaking I take the medication then I get his dinner. | Male, 66 years, N.10, TIA<br><br>Male, 68 years, N.09, TIA | A female survivor described keeping the pill box in a specific location in the house, such as by the kettle, which then acted as a reminder to check the medication box.<br>A survivor suggested using a white board and having method in place helped. She remembered taking her own medications through repetition or linking tablet use to another everyday activity. | Female, age 60 years, age at stroke 60 years,<br>Female, age 54 years, age at stroke 46 years, N.19 |
| **9. Changing medications** | They changed his medication to cheaper cholesterol and Dean was physically ill. He couldn't cope on it at all so he went back and the doctor said 'oh well it was just to try and they put him back on the others.<br>She gave me an extra pill and I had a horrific night. She made an apology and said I'm sorry it took so long to get it right, but the fine tuning takes a bit of doing. | Female, caregiver, N.24, age unknown<br><br>Male, 80 years, N.04, TIA | A survivor described being on 80 mg of simvastatin which they were happy with but that upon leaving hospital the dose was halved by the consultant which had very bad consequences, resulting in daily angina turns for a week. In the end he had to go back to his GP and be put back on the 80 mg dose.<br>A male survivor said he was taking up to 7 different blood pressure tablets and that it was unusual to only need a few tablets. He recommended going back to the GP as necessary to keep changing tablets until the correct combination was found. | Female, age 53 years, age at stroke 50 years, N.60<br>Male, age 52 years, age at stroke 52 years, N.64 |
| **10. Regimen complexity and burden of treatment** | I have to take 10 a day now altogether but I went up there (to the practice) to say can I get off some of these tablets, and I come back and I was on an extra one so I've not been up since.<br>I've got yards of them. I don't know half the names I'm just told when to take them. That's one thing I'd like to do away with. | Male, 70 years, N.13, TIA<br><br>Male, 73 years, N.11, ischaemic stroke | *A caregiver (son) was asking advice on how to encourage medication taking. His mother was originally taking multiple tablets up to 4 times a day but that now she was refusing to take them all and he was upset by this. Persuading her to continue taking the most important tablets had taken hours to do.*<br>*A caregiver (wife) described how her husband was adamant he was not prepared to take statins because he didn't have the time to keep going back to the GP for checkups. The caregiver said she was feeling helpless and wasn't sure what she could do about it.*<br>*(Burden of side effects on stroke survivor)*<br>*A survivor described similar side effects from 3 different statins despite varying the medication dosage. She said tests confirmed this and she concluded long term use could result in problems that had a negative effect on her quality of life.* | Female, age 77 years, age at stroke 77 years, N.9<br>Male, age 55 years, age at stroke 55 years, N.24<br>Female, age 34 years, age at stroke 32 years, N.36 |

Quotes in italics refer to additional themes identified in the online forum only.

*Quote transcribed.

†Quote not transcribed – described to protect user confidentiality.

‡Demographic characteristics relate to the stroke survivors only (either talking in first person or talked about by a caregiver).

GP, General Practitioner; TIA, Transient ischaemic attack.

and important insights around the practicalities of medication taking, including difficult experiences with practical aspects of 'taking medications' such as experiencing 'swallow panic'. The collaborative discussions between survivors and caregivers on the forum meant that users were likely to offer each other practical medication-taking strategies such as using a whiteboard. Caregivers in the online forum could communicate with other online caregivers separately from stroke survivors, manifesting their own opinions and attitudes towards secondary prevention medications. This did not emerge from interviews, when caregivers and survivors were interviewed together.

## DISCUSSION

In this analysis, themes that emerged from an interview study with stroke survivors, and their caregivers could be matched with corresponding themes from users of an online stroke forum. This was true despite key differences in the attributes of data collection and the lack of verification of participants' identity and stroke diagnosis. An online stroke forum can be considered a trustworthy source of data for qualitative research on patients' and caregivers' issues with medications after stroke. Perhaps because of the inclusion of a younger and computer literate population and the opportunity of online discussions between survivors and caregivers, forum data offered additional insights such as the effect of bad press on taking medicines, issues about clinician prescribing and easy access to caregivers' reflections on their caregiving role.

### Strengths and limitations

This investigation compares results from two studies addressing the same research question using two different data sources, a traditional one (interviews) and a novel one (online forum), according to the PAPA framework. The results suggest that qualitative studies on online stroke forums are strong and represent a step towards confirming that an online stroke forum is a trustworthy source of qualitative research data. A further strength is the development and use of a structured framework informed by previous literature, identifying important differences in the attributes of each of the methods of data collection. Furthermore, these results highlight characteristics that researchers could use to decide which source of data is more suitable to a particular research question, for example, an online stroke forum could be more suitable when the focus is on gathering qualitative data from young computer-literate stroke survivors and young caregivers (most forum caregivers are sons and daughters of stroke survivors). With online comments provided directly from participants, the potential for ambiguity or distortion of patient views through transcription is reduced. This investigation suggests that there is potential for an online forum to add depth to understanding of stroke survivors' issues with medications. Themes identified from the online forum matched those

that emerged from the qualitative interviews, suggesting that an online forum may well complement traditional data collection techniques for qualitative research.

Limitations of this research should also be acknowledged. While exploring barriers to medication adherence was the objective of the interview study, the online forum was not set up to investigate medication adherence and participants would not have focused their conversations on barriers to adherence to secondary prevention medication. Younger forum participants were potentially more engaged with managing their condition than their offline counterparts, perhaps representing people with a good understanding of their health.

While interviewed participants contributed to one or more themes and were included in the interview study, several forum participants mentioning secondary prevention medications were excluded from the forum study, because they did not provide enough details to allow the identification of a theme. Despite this, because of the wealth of information shared online and the high number of forum participants, the exclusion of several participants did not affect data collection within the forum study.

### Comparisons with existing research

In agreement with our findings, investigations of online versus face-to-face focus group discussions concluded that both methods could be used to answer research questions, that online forum is more suited to communicate opinions and capturing participants' perspectives from a wide geographical area[34] and, through anonymity, discussions of more personal issues.[44]

Issues around the potential reliability of forum data in answering the research question may also arise out of concerns about whether the data are skewed toward a specific participant group. As the use of the internet to conduct behavioural research grows, the representativeness of participants' samples and issues will remain challenging, in spite of the advantages of increased accessibility to otherwise hidden populations.[20]

An online forum as source of data collection offers the opportunity for cross-communication and shared support among participants. Through this forum, users can also draw on a personalised support system based on peer experiences and built trust.[45] At the same time, as shown in this study, online cross-communication between participants can enhance understanding and add depth to the themes in qualitative research. With evidence that trust forms and develops on the online forum,[46] there is the potential for this methodology to become an accepted and valued source of health information.[47]

Use of an online forum as a data collection technique in healthcare research raises potential ethical concerns around anonymity, privacy, confidentiality and informed consent compared with the more traditional qualitative approaches such as semistructured interviews.[41 48]

Compared with face-to-face interviews where stroke survivors and caregivers were interviewed together, the

forum offered an environment in which caregivers had the freedom to participate on their own (despite the existence of 'forums norms', ie, codes of etiquette or accepted topics of conversations of forums).[49] In this context, participants may be more willing to express deeply held personal opinions and to discuss sensitive issues more freely, as described by Allen and colleagues.[5] The knowledge of such issues has the potential to inform and improve involvement of both patients and their caregivers in the decision-making process, thus facilitating a collaborative approach around the use of medication and encouraging effective medication taking behaviour.[50] An interesting observation was that interview survivors were more likely to follow GPs' instructions around medicines, whereas forum survivors reported a shared approach to decisions. This identifies an interesting dynamic around how older and younger survivors view the practitioner role. Indeed, previous research confirms that older patients look to the GP for support and view the practitioner as trustworthy and an ally in making healthcare decisions.[51]

### Implications for research

The online forum represents a source of data collection suited to capturing the views of a younger stroke population who have access to online resources and to information from press outlets, which can potentially influence their attitudes to medications. The presence of younger patients and caregivers in online stroke forums offer insight for the development of interventions targeted to these groups. Indeed, research has shown that those who are younger with poorer mobility report most unmet needs, including in respect of medication taking.[52 53]

Stroke survivors who struggle with face-to-face communication but can communicate using technology such as a computer or mobile phone can provide insight on their needs, informing clinical interventions designed to improve medication taking in this patient group.

In agreement with the work on cancer forums, our work shows that the potential of online communities as a source of data is only beginning to be realised. Our findings suggest that data collected through an online forum complement traditional qualitative data collection techniques such as face-to-face interviews, giving researchers more confidence in using data from online forums. Online forum data also offered unprecedented ease of access to the caregiver perspective and the dynamic of their relationship with the stroke survivors in respect to barriers and facilitators to adherence to secondary prevention medications for stroke.

### CONCLUSION

Both interviews and online forums are rich and useful sources of data and knowledge, revealing similar issues about patients' core experience. In uncovering additional themes, the online forum may represent an important adjunct to traditional qualitative data collection methodologies.

**Correction notice** This article has been corrected since it was first published. The Acknowledgements statement has been added into the article.

**Acknowledgements** We thank Jenni Burt and ChantalBalasooriya-Smeekens for their helpful comments on the manuscript.

**Contributors** JJ contributed to the study design, data collection, data analysis and prepared the manuscript for submission. JM and SS are coinvestigators on the study and commented on the manuscript. ADS contributed to the study design, data analysis and commented on the manuscript. All authors agreed on the final draft of the submitted manuscript.

**Funding** ADS is funded by a NIHR Academic Clinical Lectureship. JJ was supported by a research grant from The Stroke Association and the British Heart Foundation: TSA BHF 2011/01.

**Competing interests** None declared.

**Patient consent** Not required.

**Ethics approval** The interview study was granted ethical approval by NHS Research South Yorkshire Ethics Committee (Ref 13-YH-0067).

**Provenance and peer review** Not commissioned; externally peer reviewed.

**Data sharing statement** No additional data are available.

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
