## [Reviewer comments · BMJ Open]

ARTICLE DETAILS

TITLE (PROVISIONAL)	An online stroke forum as source of data for qualitative research: insights from a comparison with patients' interviews
AUTHORS	JAMISON, JAMES; Sutton, Stephen; Mant, Jonathan; De Simoni, Anna

VERSION 1 – REVIEW

REVIEWER	Anneliese Synnot La Trobe University, Australia
REVIEW RETURNED	20-Nov-2017

GENERAL COMMENTS	Thanks for the opportunity to review this paper, which I enjoyed reading. I think this paper makes a really nice addition to the growing methods literature in this area; by providing a deeper analysis of the differences in online versus face to face qualitative research, and offering to structured way to systematically explore the differences between data collection approaches, which may be of use to others. I do seem to have made quite a number of suggestions (I hope they are not too onerous!), which reflects the degree to which I engaged with this paper and hope to see it further strengthened before publication. Overall, I would like to see a little more clarity and internal consistency in places (particularly around the objective), and suggest the authors highlight better/showcase their useful contribution of the attribute domains table/framework for others undertaking methods research in this area. I also offer some alternate suggestions around strengths and limitations and explanations for attribute differences that the authors may wish to consider. Abstract Objective: given this is a methodological paper, perhaps it would make more sense to lead the objective with your methods objective, not with the objective of your original research? i.e. To determine the appropriateness of an online form as a source of data for qualitative research (compared with face-to-face interviews) on medicine taking after stroke? This likely needs more tweaking but leading with the primary research aim in this sentence is confusing for the reader. I also note that this doesn't match how you have worded the objective (in fact you don't have an explicit objective) within the manuscript itself, so I think the objective needs revisiting from a consistency perspective too and amended in the manuscript. Design: You mention that this was a comparison of two 'independent' studies. What is not clear from later reading is how independent they really were. If they are conducted by the same team and analysed together or sequentially, would you really say
--

they were independent? Some more information about this within the methods of the manuscript would be helpful.

Design: It looks to me like you have two core sections in the results – (1) a comparison of themes generated by the PAPA framework and (2) a comparison of the attributes using the framework you devised. Suggest both should be mentioned here.

Results: I think it would be helpful to include the range of participant ages. The online form group is obviously younger but the ranges show they almost don't overlap in age. I think it's also an important finding that the caregivers included many more sons and daughters in the online form versus the interviews, if you can find the words to add this in (perhaps re-working the objective will help you find a few more words).

Results: You mention here the three themes that were different between online forum and interview participants yet you haven't highlighted any of your attribute findings which actually make up a large chunk of your results section. Could you find a few words to add some of your key findings here into the results? I also couldn't find that you explicitly describe these three thematic differences in the results section of your review, so it seems unusual to highlight them here and not in the manuscript itself.

Strengths and limitations

I'm not entirely sure about most of your strengths and limitations here. The first one seems like over-reach. As you highlight by citing previous literature, this is not the first methodological study of online versus face to face methods in people with a chronic condition, and the fact that it's the first one in the specific topic of B's and F's for adherence to stroke is somewhat irrelevant to me, for a methods paper. Suggest you delete this one.

The remaining dot points read more like results not me, or implications for practice, not strengths or limitations in your study itself. The one dot point that you frame as a limitation, to me is not really a limitation of your approach, but a limitation of online forums, in general. So I am not sure about the value of including any of these as strengths and limitations, either!

I suggest you could instead describe your development and use of a structured framework, informed by the literature, to systematically consider differences between attributes of data collection methods as a strength of this work.

I wonder if a limitation might be the fact that the online forum was not actually set up to investigate the phenomenon being studied.

Some of the differences you describe in attributes (i.e. people offered each other more support in the online forum compared with interviews) might not have been present if you are talking about an online forum set up by a research group to explore a particular phenomenon, as opposed to an online forum that (I assume) was specifically set up to facilitate peer support. This could also be explored in the discussion.

To find others, perhaps you could reflect a bit more on your methodological decisions made during this study to generate other strengths and limitations.

Introduction

This reads well and provides a nice intro to the paper. Just some minor points.

The 2nd para (on internet use), line two perhaps has too many ideas in the one sentence. Suggest you describe overall internet use separately to those who are older and use the internet.

I suggest a little re-working of the 7th para (page 5, line 26). The

opening line is not entirely true, there is an ever growing body of literature exploring online versus face-to-face qualitative research (much of which you have cited in this paper). To my mind, we've now got the basics (i.e. online research methods are a legitimate data collection method) and we've got a consistent body of literature about the pros and cons of each method. I think you highlight two important research questions/gaps in this section that I suggest you could frame as the gaps you are seeking to fill with this paper. These are: how we might decide whether to use one approach over another (your point in line 38) and the fact that online versus face to face is more frequently compared with respect to focus groups (your point in line 48).

I much prefer your description of what is effectively your objective (though not explicitly worded as such) at the end of the introduction (from line 52), i.e. seeking to understand the attributes that underpin each data source. Although I think you should mention your additional aim which was to compare themes generated (was it not?).

Methods

Design: Pg 6, line 17, 'Differences and similarities in the data were examined....' Introduces another potential objective/aim and seems inconsistent with earlier descriptions of the objective.

A general comment here is that it would be helpful to have a little more information about each of your primary studies within this paper (if you have the word count) rather than the reader having to go off and find the references.

Specifically, with respect to the online forum, could you clarify what was the objective of the online forum (presumably for general peer support and information sharing between stroke survivors and caregivers). You later mention in the manuscript that it was not moderated (which would be helpful in this section), but some more information about its purpose (?for information sharing, peer support etc) and how it was run would be helpful. Related to this, could you explain why you thought this would be a suitable data source for a study of barriers and facilitators to adherence to secondary prevention medication after stroke? Was there any particular focus on this when the forum was set up?

Could you also provide a little more detail about who did the analysis and when it occurred? And were the people involved in the interviews doing the analysis as well? It seems important to describe if the same researchers analysed both datasets and whether this was done concurrently or sequentially. I ask this because in my paper on a similar topic (Synnot – you cite it in your reference list), we suggested that analysing focus group data first could have 'primed' us to interpret the online focus group data, hence explaining some similarity in generation of themes. So I think its important to provide some of this detail.

Within the 'interviews-dataset' and 'online-forum dataset' sections you first describe a little of the study, and then the analysis approach. Would the analysis paragraph in each of these sections be better suited the Procedure and Analysis section?

Procedure and Analysis: I wonder if it would make this paper a little clearer if you set your two different analyses, (1) comparison of attributes and (2) the PAPA thematic analysis and comparison of themes.

Comparison of attributes framework: This is a really interesting and useful framework. Can you describe the development process in a little more detail? Did you systematically search the literature? What sort of process did you use to come to consensus within the team about the three domains and 10 attributes? Did you pilot test it and

refine further? And then I suggest you give it and the fact that you assessed attributes equal prominence in the Abstract, along with the comparison of themes.

Table 1: Sometimes its not entirely clear how you define or operationalise your attributes. For example, Participation, Response contribution, Activities and Reporting are a few attributes where I am not entirely clear what the labels mean. Did you have agreed definitions of what these attributes mean to be able to complete them for each data collection method? Perhaps you could add them as a footnote to Table 1? I am also not entirely clear what the term 'Literature (shaded) versus present study' means in the top row, i.e. you did not come up with this? It is a recognised broader feature or difference in online versus face to face research? Some clarification would be helpful.

Also, could you describe the PAPA framework with another line or two, for readers who are unfamiliar with it?

Results

Under participant characteristics could you add the age range for participants, not just the median? This seems important given the ranges almost do not overlap.

It was surprising to read so little about the comparison of themes, when you have so much (really helpful) information describing and supporting the themes generated in the PAPA framework in both studies in Table 2. As noted earlier, you don't even mention the three additional themes that were generated in the online forum. Could you add a paragraph describing these?

The section in which you discuss how the attributes related to the themes generated (p.9, line 21) is admirable, but sometimes I am not convinced about your explanations for why this might be the case. I wonder if these might be better postulated as potential explanations, rather than as evidence to support the fact that the attribute is linked with the particular difference in the theme(s). For example, the fact that you suggest the reduced geographical area of interview participants (compared with the broader sampling of the online forum) was reflected in the fact that online forum participants provided more negative views about statins (p.9, line 27) doesn't strike me as a particularly plausible explanation for this difference. To me, a more obvious explanation for this difference is that online forum participants were a lot younger, included more caregivers, particularly sons and daughters, and were more likely to be online and viewing negative press about statins. This has nothing to do with geographical scope. Another possible explanation is that carers can self-sensor when in front of their family member with a condition (as you highlight in the discussion).

You also talk about the fact that the online forum allowed more interaction between participants and sharing of advice between and within patient and caregiver groups (p.20 line 12). Sorry to cite my own research again (only for lack of time to look up other people's work – please don't feel the need to cite it again in your own paper!) but we (and I'm sure many others) found that face to face focus groups allow for people to share and learn from each other, so I'm not sure this is unique to the online environment, so much as getting a group of people with a common interest together.

I also wonder if the 'frank admissions' people made in the online forums (p.10 line 22) could've related to the younger age demographic, in addition to the fact that the interviews were conducted with the stroke survivor and the caregiver (where self-censorship could be an issue).

Discussion

	You suggest that gaining caregivers' reflections on their caregiving role (p11 line 24) may be outside the reach of traditional qualitative data sources like interviews. I think this might be over-reaching a little. There is plenty of in-depth qual research conducted with caregivers on their reflections on the caregiving role (which also doesn't really seem like it was an aim of your primary research anyway). If you had conducted the interviews with stroke survivors and carers separately, you may have got more candid caregiver reflections. The online forum also included more caregivers who were sons and daughters of stroke survivors, perhaps because they were younger and more likely to be online. Another reason could be because sons and daughters may not self-identify as a caregivers, or may not in fact, have been the primary caregiver, so if you sought to recruit people in your interviews who were stroke survivors and their caregivers, it is perhaps not surprising that you got more caregivers online. You describe your results as moving towards establishing validity (p11 line 42) and later, reliability (p.12 line 43) of online forums as a source of qualitative data. Given these terms are not usually used to describe qualitative research, would it perhaps be more helpful to use some of the terminology associated with 'trustworthiness' of qualitative research (i.e. credibility, transferability; dependability; and confirmability)? I'm not sure about your limitations section (p.12. line 4) which, I note, also seem to differ from the limitations you describe in the opening section on strengths and limitations (I would've thought they should be the same). The fact that you couldn't compare differences in characteristics like type of stroke, social class and geographic location don't seem like critical demographic factors to consider (you got what I would see as the main ones, like age and survivor vs caregiver, time since stroke), so I don't see this as a limitation. The second limitation – no body language, facial expressions in online forums is not a limitation of your research, but a limitation of online forums in general. I don't think this belongs here. Again the comment about third party moderation does not seem like a limitation of your research either. See my earlier comment (under Strengths and limitations) about other potential limitations you might consider reflecting on here.
--	---

REVIEWER	Chris Allen University of Portsmouth United Kingdom
REVIEW RETURNED	04-Dec-2017

GENERAL COMMENTS	Thank you for asking me to review this paper. It is well written and I enjoyed reading it. This is an interesting and important study that looks at validating the use of online communities (in this case an online stroke forum) as a source of data for qualitative research. Use of online qualitative data is quite common in research looking at online communities. This paper though makes a substantial contribution to the research literature by attempting to validate the use of such data through comparisons with more traditional qualitative interviewing data collection methods. I struggled to find any real discussion around your ethical
--

	considerations- specifically consent for those interviewed and the data collected online. I feel this needs to be made explicitly clear in the current manuscript. I accept that you have referenced earlier papers which discussed these ethical considerations- I still think you need to at least summarise these here. I am guessing that this was an oversight that will be easily addressed with a sentence or two providing details of approval from an ethics review board etc. I also think in the limitations section you need to make reference to the fact that those participating in online communities are likely to be considerably more engaged/activated with the management of their condition than those not using the forum. This I think is especially important, because your offline data would have had the potential to include less engaged patients- who are arguably harder to reach, these patients are unlikely to feature in the dataset provided by an online community. With the online data- you are therefore possibly only able to see the practicalities and perceptions of those who probably have a good understanding of what they are taking. Those with less health literacy are therefore probably absent in the online discussions. I do think there is an over selling of the online data as a suitable substitute for offline data. I would suggest they are still very different types of data- with the offline discourse being much more subject to social desirability bias than that of online communication. This is particularly apparent in table 3 which discusses knowledge about medications and stroke. Can we safely understand peoples perceptions of medications using online data? Table 3 for example suggests that people are perhaps more cynical- whereas offline, being interview by a health care professional, you have seen a very different response. Both feature discussions around the knowledge medications and stroke, but your data suggests that this knowledge presents very differently between the two methods. I would suggest the online data is a compliment, rather than a substitute, especially given the very different demographics.
--	---

REVIEWER	Ellen Brady N/A
REVIEW RETURNED	07-Dec-2017

GENERAL COMMENTS	An important and well-structured article on comparisons between interviews and forums as research data. I would suggest, however, that rather than the two sources being comparable, they offer different insights and should be utilised with the research question in mind. Some of the unique aspects of forums (the richness of the data, discussions between forum members, ability to look at longitudinal data) also provide insights that will make them more suitable for certain questions over interviews. This is alluded to within the paper but could be brought out more strongly. Some additional comments below.  • Sensitively handling of ethical issues, with no direct quotes utilised from forum users. • While the methodology around discerning patients from caregivers and the characterisations of forums users are described in detail in additional articles (De Simoni et al., 2016; Jamison et al., 2017), aspects of this could be brought in to the current paper. The
---

	analysis of the forum posts, in particular, is unclear.  • Page 9, “Despite the inability to ask clarification questions to participants of the forum” – I am unsure what this quote refers to, is it implying that the forum users themselves could not ask follow up questions (which does not appear to be the case) or it is referring to the fact that the interviewers could not ask follow up questions? • Rather than framing forums as a neutral venue (page 13), The idea of forums ‘norms’ could be acknowledged, e.g. codes of etiquette or accepted topics of conversations of forums (e.g. Armstrong et al., 2012 - “Oh dear, should I really be saying that on here?”: Issues of identity and authority in an online diabetes community; Mudry & Strong, 2013 - Doing recovery online; Brady et al., 2016 - “You get to know the people and whether they’re talking sense or not”: Negotiating trust on health-related forums).
--	---

VERSION 1 – AUTHOR RESPONSE

Editors comments:

- Please include a statement relating to the ethical approval obtained for the two individual studies described within this manuscript, including the full names of the ethics committees that granted approval.

Response: We have now included a paragraph in the manuscript outlining the ethical process for the study on the online forum, see page 7, line 13, and a sentence confirming ethical approval for the interview study. See page 6, line 21.

- Please revise the ‘Strengths and limitations’ section of your manuscript. This section should relate specifically to the methods, and should not include a general summary of, or the results of, the study.

Response: The ‘Strengths and Limitations’ section of the manuscript has been revised accordingly, see page 3, line 13 and page 12 line 19.

Reviewer(s)' Comments to Author:

Reviewer: 1

Reviewer Name: Anneliese Synnot

Institution and Country: La Trobe University, Australia

Please state any competing interests or state ‘None declared’: None declared

Please leave your comments for the authors below Thanks for the opportunity to review this paper, which I enjoyed reading. I think this paper makes a really nice addition to the growing methods literature in this area; by providing a deeper analysis of the differences in online versus face to face qualitative research, and offering to structured way to systematically explore the differences between data collection approaches, which may be of use to others.

I do seem to have made quite a number of suggestions (I hope they are not too onerous!), which reflects the degree to which I engaged with this paper and hope to see it further strengthened before publication. Overall, I would like to see a little more clarity and internal consistency in places (particularly around the objective), and suggest the authors highlight better/showcase their useful contribution of the attribute domains table/framework for others undertaking methods research in this

area. I also offer some alternate suggestions around strengths and limitations and explanations for attribute differences that the authors may wish to consider.

Abstract

Objective: given this is a methodological paper, perhaps it would make more sense to lead the objective with your methods objective, not with the objective of your original research? i.e. To determine the appropriateness of an online form as a source of data for qualitative research (compared with face-to-face interviews) on medicine taking after stroke? This likely needs more tweaking but leading with the primary research aim in this sentence is confusing for the reader. I also note that this doesn't match how you have worded the objective (in fact you don't have an explicit objective) within the manuscript itself, so I think the objective needs revisiting from a consistency perspective too and amended in the manuscript.

Response: Thank you for this suggestion. We have changed the objective section in the abstract as suggested. See page 2, line 4.

We have also amended the description of the study objectives within the manuscript, see page 5, line 23

Design: You mention that this was a comparison of two 'independent' studies. What is not clear from later reading is how independent they really were. If they are conducted by the same team and analysed together or sequentially, would you really say they were independent? Some more information about this within the methods of the manuscript would be helpful.

Response: This is an important point and in the methods section of the manuscript we have now added more details. See page 6 line 7.

Design: It looks to me like you have two core sections in the results – (1) a comparison of themes generated by the PAPA framework and (2) a comparison of the attributes using the framework you devised. Suggest both should be mentioned here.

Response: Thank you. We have amended the design section in the abstract accordingly. See page 2, lines 7-9.

Results: I think it would be helpful to include the range of participant ages. The online form group is obviously younger but the ranges show they almost don't overlap in age. I think its also an important finding that the caregivers included many more sons and daughters in the online form versus the interviews, if you can find the words to add this in (perhaps re-working the objective will help you find a few more words).

Response: Thanks for this suggestion. We have now included the age ranges in the Results section of the Abstract. In addition, we have included a sentence highlighting that caregivers in the forum were predominantly sons and daughters of stroke survivors. See page 2, lines 15-17

Results: You mention here the three themes that were different between online forum and interview participants yet you haven't highlighted any of your attribute findings which actually make up a large chunk of your results section. Could you find a few words to add some of your key findings here into the results? I also couldn't find that you explicitly describe these three thematic differences in the results section of your review, so it seems unusual to highlight them here and not in the manuscript itself.

Response: Thank you for this important suggestion. We have now rearranged the result section to include both sets of key findings. See page 2, lines 19-20. As recommended, we have also included a description of the 3 additional themes emerged from the online forum within the results section of the main manuscript. See page 10, line 1-6.

Strengths and limitations

I'm not entirely sure about most of your strengths and limitations here. The first one seems like over-reach. As you highlight by citing previous literature, this is not the first methodological study of online versus face to face methods in people with a chronic condition, and the fact that it's the first one in the specific topic of B's and F's for adherence to stroke is somewhat irrelevant to me, for a methods paper. Suggest you delete this one.

Response: Thank you for this observation. We have deleted the first point. The Strengths and Limitations section of the manuscript has been re-written. See page 3 line 13-26.

The remaining dot points read more like results not me, or implications for practice, not strengths or limitations in your study itself. The one dot point that you frame as a limitation, to me is not really a limitation of your approach, but a limitation of online forums, in general. So I am not sure about the value of including any of these as strengths and limitations, either!

Response: We have deleted the point describing the results of the comparison in the two data sources, as we agree that it does not fit well. However, we have left the two points about the study limitations, as part of our aim is to describe the characteristics of an online stroke forum as source of data for qualitative research, as also specified in the study title. See page 3, line 22.

I suggest you could instead describe your development and use of a structured framework, informed by the literature, to systematically consider differences between attributes of data collection methods as a strength of this work.

Response: We have now included this description, see section page 3 line 13-15.

I wonder if a limitation might be the fact that the online forum was not actually set up to investigate the phenomenon being studied. Some of the differences you describe in attributes (i.e. people offered each other more support in the online forum compared with interviews) might not have been present if you are talking about an online forum set up by a research group to explore a particular phenomenon, as opposed to an online forum that (I assume) was specifically set up to facilitate peer support. This could also be explored in the discussion.

To find others, perhaps you could reflect a bit more on your methodological decisions made during this study to generate other strengths and limitations.

Response: We believe this is addressed in the last point, see page 3 line 13-15.

Introduction

This reads well and provides a nice intro to the paper. Just some minor points.

The 2nd para (on internet use), line two perhaps has too many ideas in the one sentence. Suggest you describe overall internet use separately to those who are older and use the internet.

Response: Thank you. We have now reworded this sentence, see page 4, line 15-16.

I suggest a little re-working of the 7th para (page 5, line 26). The opening line is not entirely true, there is an ever growing body of literature exploring online versus face-to-face qualitative research (much of which you have cited in this paper). To my mind, we've now got the basics (i.e. online research methods are a legitimate data collection method) and we've got a consistent body of literature about the pros and cons of each method. I think you highlight two important research questions/gaps in this section that I suggest you could frame as the gaps you are seeking to fill with this paper. These are: how we might decide whether to use one approach over another (your point in line 38) and the fact that online versus face to face is more frequently compared with respect to focus groups (your point in line 48).

Response: We have reworded the opening line of paragraph 7 to reflect your suggestion, see page 5 line 9.

We have also added an additional sentence into the manuscript which frames the gaps in the literature that this paper is seeking to address. See page 5 line 27.

I much prefer your description of what is effectively your objective (though not explicitly worded as such) at the end of the introduction (from line 52), i.e. seeking to understand the attributes that underpin each data source. Although I think you should mention your additional aim which was to compare themes generated (was it not?).

Response: We have reworded the objective in the manuscript at the end of the Introduction section. See page 5, line 23. We have also mentioned the additional aim of comparing the themes generated in the online forum and qualitative interviews. See page 2, line 8

Methods

Design: Pg 6, line 17, 'Differences and similarities in the data were examined....' Introduces another potential objective/aim and seems inconsistent with earlier descriptions of the objective.

A general comment here is that it would be helpful to have a little more information about each of your primary studies within this paper (if you have the word count) rather than the reader having to go off and find the references.

Response: Thank you for this suggestion. We have now included additional information on each of the primary studies in this manuscript. See page 6 line 16 and page 7 line 4

Specifically, with respect to the online forum, could you clarify what was the objective of the online forum (presumably for general peer support and information sharing between stroke survivors and caregivers). You later mention in the manuscript that it was not moderated (which would be helpful in this section), but some more information about its purpose (?for information sharing, peer support etc) and how it was run would be helpful. Related to this, could you explain why you thought this would be a suitable data source for a study of barriers and facilitators to adherence to secondary prevention medication after stroke? Was there any particular focus on this when the forum was set up?

Response: Thank you for these useful suggestions. We have now included additional information in the manuscript about the forum study and its methodology, including its purpose and how it was delivered. See page 7, line 1-12.

Could you also provide a little more detail about who did the analysis and when it occurred? And were the people involved in the interviews doing the analysis as well? It seems important to describe if the same researchers analysed both datasets and whether this was done concurrently or sequentially. I ask this because in my paper on a similar topic (Synnot – you cite it in your reference list), we suggested that analysing focus group data first could have 'primed' us to interpret the online focus group data, hence explaining some similarity in generation of themes. So I think it's important to provide some of this detail.

Response: We have now provided additional detail about the analysis of the data in both primary studies including details of how analysis was undertaken and who conducted the analysis. See page 6 line 7-9.

Within the 'interviews-dataset' and 'online-forum dataset' sections you first describe a little of the study, and then the analysis approach. Would the analysis paragraph in each of these sections be better suited the Procedure and Analysis section?

Response: This is a fair point. Accordingly, we have moved the description of the re-analysis of the interview data according to PAPA to the beginning of the 'Procedure and analysis' section, as this was part of the work we have undertaken here, see page 7, lines 24-26. We opted to leave the details of the analyses of the previous two studies under each dataset description, to aid clarity. See page 6, line 15 and line 25

Procedure and Analysis: I wonder if it would make this paper a little clearer if you set your two different analyses, (1) comparison of attributes and (2) the PAPA thematic analysis and comparison of themes.

Comparison of attributes framework: This is a really interesting and useful framework. Can you describe the development process in a little more detail? Did you systematically search the literature? What sort of process did you use to come to consensus within the team about the three domains and 10 attributes? Did you pilot test it and refine further? And then I suggest you give it and the fact that you assessed attributes equal prominence in the Abstract, along with the comparison of themes.

Response: The authors have described in greater detail the development process of the attributes framework used in the manuscript. See page 8, line 1-7. We have also provided additional details about the comparison of themes in the abstract. See page 2, line 7.

Table 1: Sometimes its not entirely clear how you define or operationalise your attributes. For example, Participation, Response contribution, Activities and Reporting are a few attributes where I am not entirely clear what the labels mean. Did you have agreed definitions of what these attributes mean to be able to complete them for each data collection method? Perhaps you could add them as a footnote to Table 1? I am also not entirely clear what the term 'Literature (shaded) versus present study' means in the top row, i.e. you did not come up with this? It is a recognised broader feature or difference in online versus face to face research? Some clarification would be helpful.

Response: Thank you for this suggestion. We have now included definitions of all of the attributes as a footnote in Table 1. See table 1 for details. We have also better clarified the use of shading in the first row in Table 1, see page 21.

Also, could you describe the PAPA framework with another line or two, for readers who are unfamiliar with it?

Response: The PAPA framework is described at page 8, line 15.

Results

Under participant characteristics could you add the age range for participants, not just the median? This seems important given the ranges almost do not overlap.

Response: We have now added the age range for participants both in the abstract at page 2, lines 15-17 and in the results see page 8 line 6, line 11 and line 12.

It was surprising to read so little about the comparison of themes, when you have so much (really helpful) information describing and supporting the themes generated in the PAPA framework in both studies in Table 2.

As noted earlier, you don't even mention the three additional themes that were generated in the online forum. Could you add a paragraph describing these?

Response: We have also included additional details in the results section of the manuscript on the comparison of themes, and details of the 3 additional themes identified in the forum. See page 10 lines 1-6

The section in which you discuss how the attributes related to the themes generated (p.9, line 21) is admirable, but sometimes I am not convinced about your explanations for why this might be the case. I wonder if these might be better postulated as potential explanations, rather than as evidence to support the fact that the attribute is linked with the particular difference in the theme(s). For example, the fact that you suggest the reduced geographical area of interview participants (compared with the broader sampling of the online forum) was reflected in the fact that online forum participants provided more negative views about statins (p.9, line 27) doesn't strike me as a particularly plausible explanation for this difference. To me, a more obvious explanation for this difference is that online forum participants were a lot younger, included more caregivers, particularly sons and daughters, and were more likely to be online and viewing negative press about statins. This has nothing to do with

geographical scope. Another possible explanation is that carers can self-censor when in front of their family member with a condition (as you highlight in the discussion).

Response: Thank you for this observation, which has made us realise that we accidentally did not separate the paragraph related to geographical location from the paragraph explaining the difference in participants' sampling, where we suggest the same points. See insertion of indent at page 13, line 14

You also talk about the fact that the online forum allowed more interaction between participants and sharing of advice between and within patient and caregiver groups (p.20 line 12). Sorry to cite my own research again (only for lack of time to look up other people's work – please don't feel the need to cite it again in your own paper!) but we (and I'm sure many others) found that face to face focus groups allow for people to share and learn from each other, so I'm not sure this is unique to the online environment, so much as getting a group of people with a common interest together.

I also wonder if the 'frank admissions' people made in the online forums (p.10 line 22) could've related to the younger age demographic, in addition to the fact that the interviews were conducted with the stroke survivor and the caregiver (where self-censorship could be an issue).

Response: Further to your suggestion we have also reworded the sentence relating to frank admissions made by users of the online forum. See page 11, line 8 and line 17-19

Discussion

You suggest that gaining caregivers' reflections on their caregiving role (p11 line 24) may be outside the reach of traditional qualitative data sources like interviews. I think this might be over-reaching a little. There is plenty of in-depth qual research conducted with caregivers on their reflections on the caregiving role (which also doesn't really seem like it was an aim of your primary research anyway). If you had conducted the interviews with stroke survivors and carers separately, you may have got more candid caregiver reflections. The online forum also included more caregivers who were sons and daughters of stroke survivors, perhaps because they were younger and more likely to be online. Another reason could be because sons and daughters may not self-identify as a caregivers, or may not in fact, have been the primary caregiver, so if you sought to recruit people in your interviews who were stroke survivors and their caregivers, it is perhaps not surprising that you got more caregivers online.

Response: Thank you for this observation, which has prompted a change in our summary of the main findings of this study, putting emphasis instead on the ease of access to caregivers' reflections through the online forum when compared with interviews. Caregivers of stroke survivors are notoriously difficult to recruit to studies separately of stroke survivors themselves, due to their intense care involvement and levels of exhaustion. See page 12, line 16

You describe your results as moving towards establishing validity (p11 line 42) and later, reliability (p.12 line 43) of online forums as a source of qualitative data. Given these terms are not usually used to describe qualitative research, would it perhaps be more helpful to use some of the terminology associated with 'trustworthiness' of qualitative research (i.e. credibility, transferability; dependability; and confirmability)?

Response: Thank you for the suggestion of this wording which has more relevance to qualitative research. We have used the terminology you suggested to describe our results See page 12, line 11 and line 22.

I'm not sure about your limitations section (p.12. line 4) which, I note, also seem to differ from the limitations you describe in the opening section on strengths and limitations (I would've thought they should be the same). The fact that you couldn't compare differences in characteristics like type of stroke, social class and geographic location don't seem like critical demographic factors to consider (you got what I would see as the main ones, like age and survivor vs caregiver, time since stroke), so I don't see this as a limitation. The second limitation – no body language, facial expressions in online

forums is not a limitation of your research, but a limitation of online forums in general. I don't think this belongs here. Again the comment about third party moderation does not seem like a limitation of your research either. See my earlier comment (under Strengths and limitations) about other potential limitations you might consider reflecting on here.

Response: Thank you for the suggestion provided. We have re-worded this section, seen page 13 line 14-19.

Reviewer: 2

Reviewer Name: Chris Allen

Institution and Country: University of Portsmouth, United Kingdom

Please state any competing interests or state 'None declared': None declared

Please leave your comments for the authors below Thank you for asking me to review this paper. It is well written and I enjoyed reading it.

This is an interesting and important study that looks at validating the use of online communities (in this case an online stroke forum) as a source of data for qualitative research. Use of online qualitative data is quite common in research looking at online communities. This paper though makes a substantial contribution to the research literature by attempting to validate the use of such data through comparisons with more traditional qualitative interviewing data collection methods.

I struggled to find any real discussion around your ethical considerations- specifically consent for those interviewed and the data collected online. I feel this needs to be made explicitly clear in the current manuscript. I accept that you have referenced earlier papers which discussed these ethical considerations- I still think you need to at least summarise these here. I am guessing that this was an oversight that will be easily addressed with a sentence or two providing details of approval from an ethics review board etc.

Response: Thank you. We have now included details of the ethical procedures that were undertaken for both of the primary studies described in the manuscript. See page 6 line 21-22 and page 7 lines 13-21 2.

I also think in the limitations section you need to make reference to the fact that those participating in online communities are likely to be considerably more engaged/activated with the management of their condition than those not using the forum. This I think is especially important, because your offline data would have had the potential to include less engaged patients- who are arguably harder to reach, these patients are unlikely to feature in the dataset provided by an online community. With the online data- you are therefore possibly only able to see the practicalities and perceptions of those who probably have a good understanding of what they are taking. Those with less health literacy are therefore probably absent in the online discussions.

Response: Thank you for both of these useful suggestions. We have highlighted both of these in the Limitations section of the manuscript. See page 13 line 17-19

I do think there is an over selling of the online data as a suitable substitute for offline data. I would suggest they are still very different types of data- with the offline discourse being much more subject to social desirability bias than that of online communication. This is particularly apparent in table 3 which discusses knowledge about medications and stroke. Can we safely understand peoples

perceptions of medications using online data? Table 3 for example suggests that people are perhaps more cynical- whereas offline, being interviewed by a health care professional, you have seen a very different response. Both feature discussions around the knowledge of medications and stroke, but your data suggests that this knowledge presents very differently between the two methods. I would suggest the online data is a compliment, rather than a substitute, especially given the very different demographics.

Response: We agree that the two sources of data are complementary to each other rather than one better than the other, see added text at page 13, lines 5-7 and Implications for research at page 15, line 19-21.

Reviewer: 3

Reviewer Name: Ellen Brady

Institution and Country: University of Manchester

Please state any competing interests or state 'None declared': None declared

Please leave your comments for the authors below. An important and well-structured article on comparisons between interviews and forums as research data. I would suggest, however, that rather than the two sources being comparable, they offer different insights and should be utilised with the research question in mind. Some of the unique aspects of forums (the richness of the data, discussions between forum members, ability to look at longitudinal data) also provide insights that will make them more suitable for certain questions over interviews. This is alluded to within the paper but could be brought out more strongly.

Response: This is in agreement with reviewer's 2 suggestions. We agree that the two sources of data are complementary to each other rather than one better than the other, see added text at page 13, lines 5-7 and Implications for research at page 15, line 19-21.

Some additional comments below.

- Sensitively handling of ethical issues, with no direct quotes utilised from forum users.
- While the methodology around discerning patients from caregivers and the characterisations of forum users are described in detail in additional articles (De Simoni et al., 2016; Jamison et al., 2017), aspects of this could be brought in to the current paper. The analysis of the forum posts, in particular, is unclear.

Response: Thank you. We have now included additional details in the manuscript describing the analysis of the forum posts. See page 7, lines 4-12.

- Page 9, "Despite the inability to ask clarification questions to participants of the forum" – I am unsure what this quote refers to, is it implying that the forum users themselves could not ask follow up questions (which does not appear to be the case) or it is referring to the fact that the interviewers could not ask follow up questions?

Response: Thank you for this observation. This refers to the inability to probe the forum participant's line of questioning. This sentence has now been re-written to clarify the point.

See page 11, line 2

- Rather than framing forums as a neutral venue (page 13), The idea of forums 'norms' could be acknowledged, e.g. codes of etiquette or accepted topics of conversations of forums (e.g. Armstrong et al., 2012 - "Oh dear, should I really be saying that on here?": Issues of identity and authority in an online diabetes community; Mudry & Strong, 2013 - Doing recovery online; Brady et

al., 2016 - "You get to know the people and whether they're talking sense or not": Negotiating trust on health-related forums).

Response: Thank you. This is an important point and we have mentioned it at page 14, line 23-24.

VERSION 2 – REVIEW

REVIEWER	Anneliese Synnot La Trobe University, Australia
REVIEW RETURNED	22-Jan-2018

GENERAL COMMENTS	Very happy for the manuscript to proceed to publication.
--

REVIEWER	Chris Allen University of Southampton
REVIEW RETURNED	27-Jan-2018

GENERAL COMMENTS	I enjoyed reading this paper and the revisions have improved it. I am hoping to use similar methods in the future as a result of reading this paper. Thank you.
---

REVIEWER	Ellen Brady University of Manchester
REVIEW RETURNED	07-Feb-2018

GENERAL COMMENTS	Thank you for your revisions. This has addressed issues around the methodology and ethics of the paper and better highlights the contribution that this research makes to the literature.
---